# Professional Virtues for a Responsible Adaptation to Sea Level Rise

Anna Wedin 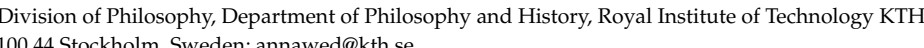

Division of Philosophy, Department of Philosophy and History, Royal Institute of Technology KTH, 100 44 Stockholm, Sweden; annawed@kth.se

**Abstract:** In the field of responsibility and climate change, much attention has been paid to actions and what we need to do in order to take responsibility. This paper shifts the perspective from what we should *do* to how we should *be* in order to be responsible. Looking at the case of local adaptation to sea level rise, the question of what characterizes a responsible planner is addressed. Departing from the idea of professional virtues, aspirational characteristics are extrapolated from three codes of ethics for planners. The identified virtues are discussed in relation to the central challenges of adaptation to sea level rise, giving an indication of which virtues are most important in this given context. When placing the responsible planner in an institutional context, it is evident that while a virtue perspective should not replace a system analysis, it provides a valuable complement to the traditional action-focused discourse on responsible adaptation.

**Keywords:** responsibility; adaptation; sea level rise; professional virtues; codes of conduct; codes of ethics; planning





## 1. Introduction

Climate change poses many challenges, not least with respect to the distribution of responsibility for mitigation, adaptation, and compensation. In the discourse on responsibility, focus tends to be on actions: we have a responsibility to do certain things. An alternative perspective does not focus on which actors should take what responsibility, but how actors should be in order to behave responsibly. In this paper, the virtues that ought to be guiding when taking responsibility are addressed, specifically looking at local planners involved in adaptation to climate change-induced sea level rise.

Historically, the academic discourse on responsibility and climate change has been focused on assigning nations responsibility based on different principles, including: the polluter pays, ability to pay, and beneficiary pays principles [1]. What has been discussed to a lesser extent is distribution of responsibility at subnational levels. Local and regional governments, as well as the private sector are important actors in addressing climate change [2,3]. This becomes especially evident as the perspective is shifted from mitigation to adaptation. Mitigation is about limiting climate change and adaptation consists of actions that are intended to decrease our vulnerability in the face of climate change. It can be understood as the practice of organizing society and the environment in such a way that our needs can be satisfied and values protected over a long time, which to a larger extent needs to be contextualized and addressed at a local level.

Responsibility for adaptation to climate change can be broken down into responsibility for initiating action, responsibility for implementing action, responsibility for paying for action, and responsibility for the residual risk [4]. Within each of these fields, there are several practical challenges, sometimes called barriers or limits to adaptation [5,6]. Among these are epistemic issues such as the fact that we do not know the extent or pace of the effects of climate change, complicating long-term planning. Adaptation to climate change is also challenged by institutional shortcomings, such as lacking frameworks for addressing

long time-horizons [7]. Facing these challenges, planners express that they need more resources and better guidelines in order to live up to their responsibility.

However, due to the complex nature of climate change and its effect, it is, if not impossible, at least difficult to formulate guiding rules and principles for all that needs to be done. Sandler states that "if action guidance cannot always be accomplished by rules and principles alone—then the wisdom and sensitivity that are part of virtue are in some situations indispensable for identifying right action" [8] (p. 8). Therefore, I suggest that in addition to trying to formulate principles and distribute responsibility justly, there is a need to look to the planners themselves and at what constitutes a responsible planner. Garrath Williams argues that responsibility should be understood as a good-making trait of character and defines it as "the readiness to respond to a plurality of normative demands" [9] (p. 469). Whether adhering to this understanding of responsibility as a virtue, or merely seeing it as a quality of character or moral disposition, there is a value in trying to be more specific when discussing what it means to be responsible in a particular context; in this case subnational adaptation to climate change.

The impacts of climate change can take many different forms, and so can adaptation to climate change. In order to focus my investigation, I will address adaptation to sea level rise induced by climate change. Sea level rise is a global concern, but adaptation to sea level rise is mostly concerned with physical planning and the built environment, and hence an issue commonly addressed by planners at a local or subnational level [10]. I am therefore concerned with what traits signifies a virtuous regional or urban planner working with adaptation to sea level rise. In order to answer this question, I begin by presenting the concept of professional virtues and explaining how the defining features of a profession calls for different virtues. Next, I turn to three codes of ethics for planners to investigate what virtues are promoted in them. Having identified these aspirational characteristics, they are discussed in relation to the particular challenges of adaptation to sea level rise. The paper ends with a discussion on the context in which the 'responsible planner' exists and suggesting ways in which the virtues can be cultivated, and a few concluding remarks.

## 2. Professional Virtues

The idea of professional virtues is inspired by classical virtue theory, which focuses on moral character in moral evaluation. That means that unlike consequential and deontological ethics, virtue ethics is less concerned with action and what we should do, but more with how we should be. While traditional virtue ethics are concerned with all domains of human life, professional virtue theory address virtues of relevance to a person's profession. The term virtue itself has its etymological roots in the Latin 'virtus', which in turn is linked to the ancient Greek term 'arête', meaning excellence. The idea of excelling is central to virtue theory and a virtue can be understood as "any stable trait that allows its possessor to excel in fulfilling its distinctive function" [11]. Professional virtues can thus be understood as those stable traits that allows a professional to excel in fulfilling the distinctive function of that profession.

Another way of explaining this is by departing from Alasdair MacIntyre's understanding of virtue ethics, which presupposes a human practice with internal goods. MacIntyre describes his conception of practice as follows:

> "By practice I am going to mean any coherent and complex form of socially established cooperative human activity through which goals internal to that form of activity are realized in the course of trying to achieve those standards of excellence which are appropriate to, and partially definitive of, that form of activity, with the result that human powers to achieve excellence, and human conceptions of the ends and goods involved, are systematically extended".

[12] (p. 187)

As we can see, MacIntyre describes a practice as a cooperative activity that is sufficiently coherent and complex. Moreover, it is stated that there are goals internal to this form of activity that are realized as we try to achieve standards of excellence definitive

of that form of activity. A profession can be defined as "a number of individuals in the same occupation voluntarily organized to earn a living by openly serving a moral ideal in a morally permissible way beyond what law, market, morality, and public opinion would otherwise require" [13]. Professions can therefore be understood as practices. Professionals often organize in professional communities and carry out their work from a common theoretical and practical base of knowledge. Other central features of professions include that they have the mandate of society to provide services in a specific field, and that they have common goals that they are striving towards [14]. Excellence in a practice relate to these goals, and as excellence is achieved, those who are achieving it and the practice itself progress, thus furthering the internal goals of the practice.

In order to engage well in practices, technical skills, knowledge and virtues are needed [15]. The technical skills and knowledge relating to particular professions are often more well-defined than the required professional virtues. According to some authors there are virtues that are important to all professionals; for example, William May has listed perseverance, courage, integrity, public spiritedness, benevolence, and humility [16], and Aiaraksinen mentions integrity, trustworthiness and responsibility [17]. However, following from the understanding that professions can be seen as practices, each with individual standards of excellences and internal goods belonging to it, these and other virtues might be more or less important for different professions. In the literature on professional virtues, central virtues for a number of professional spheres have been identified, including nursing [18], teaching [19], journalism [20], engineering [21], and business [22]. So far, no one has looked at planning for adaptation to sea level rise from a professional virtue perspective. A more thorough investigation of virtues in the context of adapting to climate change has, however, been called for. Thompson and Bendik-Keymer write: "who we are today is not ready for this mess and who we have been got us into this mess" [23] (p. 15, author's emphasis), proposing that we change our general conception of humanity and goodness. I wish to make a more specific contribution, focusing on professional virtues of those working with adaptation to sea level rise.

Adaptation to sea level rise will largely take the form of physical planning, be it to build walls to protect society against more frequent floods or managed realignment, meaning moving people and property away from at-risk zones. The mission and domain of planning has been proposed as "Planning to Enhance the Quality and Life for All in the Built Environment" [14]. Since planning has a shared moral ideal uniting people in a shared occupation, planning should be thought of as a profession. Concretely, planning can be understood as "the formulation, content, and implementation of spatial public policies" [24] (p. 395). However, formulating and implementing adaptation policy is complicated as adaptation requires planning for unusually long time-horizons, and is surrounded by uncertainty on both the extent and pace of sea level rise [25]. Adaptation therefore involves making decisions on risk exposure, e.g., by building in an area that might or might not be flooded in the future. Athanassoulis and Ross suggest that in these situations, we should not morally judge decisions on risk based on what happens—since due to the very nature of risk, this is beyond our control [26]. Rather, they propose, our focus should be on the time when a decision is taken and how reasonable the actor is at this stage. The importance of the virtue practical wisdom (phronesis), which was originally discussed by Aristotle, is highlighted by the authors. Practical wisdom is the virtuous person's ability to judge the relative relevance of factors pertaining to a situation and weigh up these features in making moral decisions. It is the ability to translate the idea of moral good into concrete or practical situations. It can thus be understood as a purely good-willed cleverness, ruling how and when virtues are to be displayed [27]. I will further elaborate on practical wisdom in the following sections. Besides practical wisdom, there are likely other virtues that planners would benefit from cultivating and displaying in the context of adaptation to sea level rise. In the next section, I will turn codes of ethics to identify some such aspirational characteristics for planners.

### 3. Extracting Virtues for Planners

Seeing planning as a practice, there are internal goods that are defining of planning. These are realized as we try to achieve certain standards of excellence, in part through cultivating and exercising virtues. In this paper, a bottom-up approach to identifying these virtues is taken. Instead of attempting to define ultimate goals of planning in order to then identify virtues that would contribute in achieving these goals, different ideals that are promoted by those currently engaged with the practice will serve as a point of departure for this investigation. In analyzing ethical codes, it is possible to get an idea of what kinds of behaviors and characteristics planners themselves promote or aspire to [28].

One might consider it contradictory to depart from professional codes when this project is motivated by the inadequateness of rules and principles in complex situations such as planning. Indeed, professional codes have been criticized on the basis that general rules seldom cover all or even most practical cases, that it promotes an unreflective or conformist mindset, and thus do not cannot capture the context sensitive deliberation of practical wisdom [29]. Here, it is important to be aware of the difference between codes of conduct and codes of ethics. The difference is that the former is concerned with following legislation and maintaining a professional conduct (such as not accepting bribes and respecting business agreements), and the latter is more visionary in promoting behavior that goes beyond this [30]. Codes of conduct are thus more concerned with doing (and even more so, not doing), while codes of ethics focus on being. It is true that when focusing on codes of conduct, we are stuck with rules that give limited guidance in unexpected or complex situations. However, aspirational ideals as presented in codes of ethics can provide an insight into what is thought to be required of a planner in order to achieve excellence in the field or practice. In order to get an understanding of what the moral and epistemic ideals of planners are, codes of ethics are therefore a suitable starting point.

In this paper, three codes of ethics for planners put forward by different organizations are used: the *Code of Professional Conduct* by the British organization Royal Town Planning Institute [31], the *Code of Ethics and Professional Conduct* by the American Institute of Certified Planners [32], and the *Code of Ethics* by the Urban Land Institute [33]. The first is an organization for planners in the UK, the second for American planners, and the third is a global organization which is somewhat more oriented towards real estate, but that also addresses land use. While they are labelled differently, they all include elements that promote ideals rather than rules and guidelines, and these are the sections that have been analyzed in the pursuit for professional virtues for planners. The three organizations represent planners from different regions around the world, and while they might not give the whole picture of what we want from a planner, the selection of studied codes at least gives an indication of what is required from planners, if not globally, at least in a Western context. In what follows, the results and analysis from a systematic reading with emphasis on aspirational ideals present in the codes are presented.

#### 3.1. Royal Town Planning Institute

The RTPI was established following the 1909 Housing and Town Planning Act and is today described as a "membership organisation and a chartered institute responsible for maintaining professional standards and accrediting world class planning courses nationally and internationally". The RTPI's *Code of Professional Conduct* emphasizes professional conduct and actions, and among the 29 points addressed in the document statements can be found, such as "Members must take all reasonable steps to ensure that their private, personal, political and financial interests do not conflict with their professional duties" and "Members must discharge their duty to their employers, clients, colleagues and others with due care and diligence". The majority concern conforming to regulation, which does not say much about, or reflect upon character. However, some of the issues raised indicate that certain characteristics are promoted. For example, the code states that "Members must act with *honesty* and *integrity* throughout their career".

The RTPI code further states that "Members must exercise fearlessly and impartially their independent professional judgement to the best of their skill and understanding". While fearless indeed could be interpreted as the vice recklessness, I propose a generous reading and interpret the code to encourage virtues such as *courage* and *perseverance*. The code also suggests that its members should "maintain professional competence", which arguably could be understood as campaigning *love of knowledge* (Love of knowledge is a virtue which was first proposed by Plato. It can be understood as the characteristic of constantly wanting to learn more), alternatively some kind of more general continuous self-improvement, which is at the very core of virtue theory. Finally, two points refer to the role of the planner to promote equality and eliminate discrimination, which is an endeavor that could be motivated by the virtue of *justice*.

### 3.2. The American Institute of Certified Planners

Second, we turn to the AICP, which is an independent organization verifying planners. It communicates a code of *Ethics and Professional Conduct*. The document is divided into five parts where the first consists of aspirational principles, the second rules of conduct and the remaining three different procedural provisions. The procedural provisions address how advisory ruling, complaints, and disciplinary action should be dealt with, and due to their practical nature, they will not be discussed further here. The principles of the other two parts derive from what the AICP calls the "special responsibility of our profession to serve the public interest with compassion for the welfare of all people and, as professionals, to our obligation to act with high integrity". The rules of conduct contain rules to which planners can be held accountable. These are presented as what one should not do, e.g., "We shall not deliberately or with reckless indifference fail to provide adequate, timely, clear and accurate information on planning issues". The aspirational principles, on the other hand, "constitute the ideals to which we are committed". This has a clear motivational aspect to it and will be the focus in this analysis.

There is a total of 21 aspirational principles. These are categorized under the headings of 'Our overall responsibility to the public', 'Our responsibility to our clients and employers', and 'Our responsibility to our profession and colleagues'. As mentioned, responsibility can be understood as a good-making characteristic or virtue. However, the purpose of this paper is to explore what kinds of virtues can be derived from an understanding of responsibility in the context of adaptation to sea level rise. Responsibility will therefore not be listed as a virtue in this paper.

As an example of the planner's overall responsibility towards the public, planners are recommended to seek social justice and deal fairly with all participants in the planning process. This can be interpreted as a need for *public spiritedness* and *justice*. They should also "pay attention to long-range consequences and be aware of the interrelatedness of decisions", which requires *practical wisdom*. When it comes to responsibility to clients and employers, it is stated that "We owe diligent, creative, and competent performance of the work we do in pursuit of our client or employer's interest". In the profession of planning, *diligence* and *creativity* can thus be understood to be important virtues. An example of "responsibility to our profession and colleagues" is that planners should not accept "the applicability of a customary solution without first establishing its appropriateness to the situation" and "continue to enhance our professional education and training". This is claimed to protect and enhance the integrity of the profession and would be helped by fostering *humility* and *love of knowledge*.

As a final comment on the AICP code, it is also recommended that planners "shall systematically and critically analyze ethical issues in the practice of planning". Noteworthy is that the document explicitly states that the aspirational values can come into conflict with each other, and that in order to reach an ethical judgement, "a conscientious balancing, based on the facts and context of a particular situation and on the precepts of the entire Code" will be required. These two statements can be understood as the requirement of planners to foster *practical wisdom*.

*3.3. The Urban Land Institute*

Third, we turn to ULI, which states its mission to be to provide leadership in the responsible use of land and in creating and sustaining thriving communities worldwide. ULI presents a code of ethics which emphasizes that the planner should show respect for a number of elements: respect for the land, the profession, the consumer, the public, equality of opportunity, others in the land use and development profession, the larger environment, the future, future generations, and personal integrity. Under respect for the profession, it states that "I will observe the highest standards of professional conduct and will seek continually to maintain and improve my professional skills and competence". As with the similar phrasings in the AICP code, this can be understood as supporting the ideals of excellence and self-improvement. In the section on respect for others, public *spiritedness* and *justice* are implicated. The concluding point, Respect for Personal Integrity, requires that:

> "I will employ the highest ethical principles and will observe the highest standards of integrity, proficiency, and honesty in my professional and personal dealings. I will remain free of compromising influences or loyalties and will exercise due diligence in ensuring that my performance is at all times creatively, competently, and responsibly managed".

This quote reflects that the planner should display *integrity* and *honesty*, and to carry out their work guided by *creativity*. Worth mentioning is that the ULI code is broader than the previous two in the sense that it emphasizes non-anthropocentric elements to be considered in the planning process. It states that respecting the land means that "I know that each parcel of land is a precious, distinct, and irreplaceable portion of this distinct and irreplaceable planet. I will treat it with the respect that it deserves, recognizing that I will be judged by the integrity and permanence of my developments, which will survive my lifetime". This suggests that planners need to consider not only the well-being of the human community they are planning for, but that they should also protect the environment for its intrinsic value. This conduct requires great capacity for ethical and practical deliberation, meaning a fostering of *practical wisdom*.

## 4. Virtues for Adaptation

The full list of identified virtues from the three codes of ethics for planners reads: honesty, integrity, courage, perseverance, love of knowledge, justice, public spiritedness, practical wisdom, diligence, creativity, and humility. From looking at three guiding documents for planners, it is impossible to say if these virtues are the only, or even the most important virtues for planners to cultivate. However, they paint a convincing image of a virtuous planner, and I will assume that they are at least of some value in achieving the internal goals of the practice that is planning.

Having identified these virtues, one might ask how it is possible for planners to make use of them in the context of adaptation to sea level rise. One premise that this investigation is built upon is that adaptation to sea level rise is related to planning. While other forms of adaptation may deal with, e.g., public health policy or agricultural strategies, adaptation to sea level rise is largely about protecting societies from a physical threat, through the construction of physical barriers or by moving buildings and infrastructure out of risk zones. Adaptation to sea level rise can thus be seen as a subcategory to planning, and a virtuous planner would be best suited for carrying it out. The identified virtues could therefore, without further analysis be said to be those needed when adapting to sea level rise. However, it is possible to further specify the practice of adaptation in order to identify virtues that are particularly useful in realizing its goals. In this section, I will address challenges of adaptation that sets it apart from planning in general and point out certain virtues as particularly valuable in this context.

First, there are epistemic challenges to adaptation that distinguishes it from traditional planning. The goal of both traditional planning and adaptation can be understood as organizing society and the built environment in such a way that our needs can be satisfied

and values protected over a long time period. While traditional planning always has needed to consider an uncertain future, this uncertainty expands in the face of climate change and sea level rise. There is uncertainty on how much and how fast sea levels will rise, making the risk of sea level rise almost impossible to calculate, and yet the decisions we take today may well influence the impact that sea level rise will have in the future. Due to the complexity of the challenge and the difficulty of predicting outcomes, it is valuable to assess how those working with adaptation respond to and reason within the adaptation process. This calls for epistemic virtues.

When I speak of epistemic virtues, I do not refer to the particular intellectual virtues proposed by Aristotle, but to the vaguer idea of characteristics that promote intellectual flourishing, or "characteristics that make us critical thinkers" [34]. It is clear that since adaptation to sea level rise is surrounded by epistemic challenges, epistemic virtues will be of importance in order to achieve excellence in this particular practice. Facing the challenge of uncertainty, the characteristics that I propose as central in the process of adapting to sea level rise are *love of knowledge* and *humility*. Love of knowledge, as the desire to learn and understand more is crucial in a situation where there is much uncertainty, in order to strive towards making as well-informed judgements as possible. For any profession which requires a complex understanding of how things work, love of knowledge drives the professional in becoming better at their job and as a person. At the same time, it is important that epistemic humility is guiding in the process of adaptation. Epistemic humility refers to having the right attitude to one's own intellectual ability and one's dependence on others intellectually [35]. Since planning is situated in the public sphere, it is crucial that planners are open to the views of others but also rely on their own knowledge. Without humility, there is a risk that planners lock themselves into established patterns and fail to be open to new perspectives. Intellectual honesty, understood as the willingness to revise and adjust one's beliefs in the face of new knowledge is a related important virtue.

Adaptation to sea level rise also poses practical challenges requiring virtues relating to the moral sphere of the professional context. For example, the effects of sea level rise are likely to affect vulnerable groups disproportionally. Moreover, not only sea level rise, but also the adaptive measures themselves, can contribute to injustice [5]. These issues are complicated by the fact that adaptation is both surrounded by large uncertainty and requires planning for time-horizons beyond what is common practice. Navigating these challenges, I propose that the virtue of *justice* ought to guide planners working with adaptation. Justice denotes excellence in determining what is due to whom [36], and should ideally be sensitive both to those currently living and those in the future who will be affected by our actions today. A just planner will likely be better fitted in taking decisions that contribute to our needs being satisfied and protected over time. Another challenge relates to the fact that the very need for adaptation is questioned by those skeptical towards the existence and severity of climate change and sea level rise, who dismiss legitimate concern as words of doomsday prophets. Even when there is an understanding that climate change is real, adaptation issues are sometimes rejected in favor of short-term projects which give more direct reward and are popular among the electorate. In these situations, planners working with adaptation will need *courage* to dare to be uncomfortable and stand by the the best avaible knowledge and climate science, even when it can come across as controversial or in extreme cases lead to retaliation.

Finally, the need to develop *practical wisdom* is crucial for those who are working with adaptation to sea level rise. Practical wisdom can be seen as ability to balance the other virtues against each other, both in the moral and epistemic domain. It is the virtuous person's ability to judge the relative relevance of factors pertaining to a situation and weigh up these features in making moral decisions. Adaptation requires careful balancing of different long-term goals and short-term priorities. It also involves accounting for many different groups and values, and navigating in political landscapes. This conclusion aligns that the analysis that a virtuous conduct to risk consists of "sensitivity to morally relevant features of situations (phronesis) and the capacity to be moved by morally relevant

considerations" [26] (p. 225). Practical wisdom thus helps balancing love of knowledge, humility, justice, and courage, which together form a profile of the character best suited to address the challenge that is adaptation to sea level rise.

## 5. The Virtuous Planner and Institutions

I will now turn to address a few anticipated concerns with this project and its conclusions regarding the context in which the virtuous planner exists. One might argue that in focusing on individuals' virtues, the pivotal role of institutions is missed. Indeed, institutions are in many ways better suited than individuals in addressing the challenges of climate change, including adaptation to sea level rise. Henry Shue has suggested that responsibility should fall on institutions rather than individuals, for reasons of efficiency and fairness [37]. He claims that only institutions can make possible the coordination and cooperation that are needed, and that it would be too much to demand of individuals to live up to climate related duties. However, institutions are made up of individuals or professionals, who work to realize the internal goals of their practice. It is evident that the professional does not easily fit into the dichotomy of individuals and institutions that has been so influential in the discussion on responsibility and climate change; this account might help addressing this problem. Even if we believe responsibility for adaptation should lie with institutions, the character of the professional is of central importance.

Having said that, the context that professionals exists within can affect how they are able to cultivate and exercise their virtues. By context, I mean the institutions that professionals are working within, which in the case of the planner often means local and regional authorities and government agencies. Since institutions can be politically controlled, corrupt, or over-loading agents working within them, it can be difficult for planners to cultivate or live up to the ideals mentioned in this paper. Bendik-Keymer states that his concern "about focusing on the 'who are we?' question is that we risk moralism and miss morality's complex relation to its organizational context and to human nature" [38] (p. 273). In order to acknowledge the complex relation of organizational contexts and morality, there is certainly a value in addressing institutional shortcomings when it comes to adaptation. A virtue perspective can help in doing this. In addition to fostering professional virtues of individuals, it has been argued that organizations themselves should also cultivate virtues [15].

More concretely, there is a need to foster appropriate institutional environments; Larry May suggests that responsibility of institutions involves facilitating and encourage a virtuous behavior [16]. Only this way, planners can cultivate and practice the virtues that will help them achieve the goals of their profession. However, May does not mention this kind of institutional culture will be achieved and maintained. There is no question that following rules and regulation will play a part in achieving a culture in which virtuous behavior is encouraged; democratic processes, the absence of corruption, and reasonable working conditions are central elements of such a culture. However, even if such a culture is achieved, this will make little difference unless the agents within the institution are virtuous. Therefore, we cannot neglect the role of the individual.

Here, we must ask, how can we promote virtuous behavior of individuals within institutions? Without offering a fully worked-out suggestion, I would like to point to that institutionalized schemes for cultivating virtues do exist, such as compassion training which is used in the health and care sectors. Sinclair et al. argue that compassion stems from virtues of genuineness, love, honesty, openness, care, authenticity, understanding, tolerance, kindness, and acceptance [39]. They explain that lack of compassion has been identified as a leading cause of failures within the health sector and that all health professionals thus should be trained in compassion. Compassion can be particularly important in dealing with "difficult patients", a significant challenge within the care profession [40]. If it is possible to train care-workers in compassion, could it not be possible to train planners in responsibility? Like compassion, responsibility stem from a number of virtues, and just as it is particularly important to be compassionate in the most challenging situations, this

is where the demands on the responsible planner is at its highest. Responsibility training could be a part of the education of planners and also be continuously be implemented within the organizations they are working. Such responsibility training could include case-driven dilemma sessions or roleplays focusing on the virtues that have been addressed in this paper.

A final potential objection is that this project could be interpreted as suggesting that planners are currently not responsible. This is not the case. Central to virtue theory is the concept of excellence, and even though planners might currently possess the identified virtues, there is always room for improvement. Being virtuous is not a 'quick-fix', but something that requires life-long engagement and training. The practice of planning is a complex cooperative exercise, and planning for adaptation is arguably even more difficult. Within this practice, it is possible to do better and in doing this, pushing the boundaries for standards of excellence belonging to planning. By moving away from the negative duties of codes of conducts towards the aspirational ideals that can be found in ethical codes, we provide planners with opportunities to become better, and further the goods of planning beyond their current state.

## 6. Conclusions

To sum up, in this paper the idea of responsibility as a virtue has been analyzed in the context of planning for adaptation to sea level rise. Adaptation to sea level rise is largely concerned with the built environment and therefore intrinsically connected to urban and regional planning. Planning is a profession and can be thought of as a practice in MacIntyre's sense, and therefore has internal goals which the cultivation and exercise of professional virtues can help to achieve. In order to identify these virtues, the study departed from three codes of ethics for planners, identifying eleven professional virtues that planners should cultivate. The identified virtues are the content that fill the term 'responsibility' in the context that is planning. The practice was then further specified to adaptation to sea level rise, and departing from the challenges surrounding that practice, five virtues stood out as most important for planners to cultivate and exercise. These are: *love of knowledge*, *humility*, *justice*, *courage*, and *practical wisdom*. Naturally, the context that the planner works within will affect the extent to which they will be able to exercise virtues. However, this is no excuse for neglecting the role of the planner.

We must not forget that institutions are made up of individuals and particularly when institutions are flawed and do not provide the guidance and frameworks that are necessary, there is a need for planners to work on their professional virtues, in order to be able to make the best out of a difficult situation. Addressing a difficult task when challenged by institutional shortcomings, a planner working with adaptation can cultivate their virtues and hopefully make a change. In fact, it is exactly because some institutions do not live up to our ideals that we need to look to individuals to be responsible and work toward being virtuous in order to do good. It might be thought of as asking too much of the individual planner, but sadly, I think that says more of the challenge we are facing. After all, it is more constructive to focus on what to do with the limited power each person holds, than to emphasize ways in which complete control is lacking [24].

For these reasons, as we face the challenge of climate change and sea level rise, a focus on who we are and want to be can truly be valuable. Taking the approach that responsibility is something that we are, and not something that we do broadens the discourse on responsibility and adaptation. As previously stated, the purpose of this paper is not to argue that such a virtue perspective should replace an action-perspective completely, but rather complement it. Furthermore, it seems like we have little to lose from adding this perspective; striving towards cultivating professional virtues will likely have the effect of self-improvement more generally, which can prove valuable in a wider context. Although professional virtues are practice- and context-dependent, the virtues we have discussed here can be generalized to other practices. They do not provide direct action guidance in the context of adaptation or beyond, but make agents possessing them

better fitted to deal with the challenges that life brings, extending human powers to achieve excellence.

**Funding:** This research was funded by FORMAS, grant number 2016–20135.

**Conflicts of Interest:** The author declares no conflict of interest.

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
