# Peer review of "Professional Virtues for a Responsible Adaptation to Sea Level Rise"

_philosophies, doi:10.3390/philosophies6020037_

Round 1
Reviewer 1 Report
This is an original piece of work; clearly presented and well argued. There are some virtue terms that could have been more rigorously explained. For example, the 'phronesis' (line 132) has a rather vague description (though its treatment later in the text gets a more robust explanation). Normally, phronesis refers to a goal/ideal or telos (the overall moral good and the ability to translate that good into concrete or practical situations); the telos here could have been more clearly identified. However, given the professional ethical context of this investigation, this is probably a minor issue. A more substantial issue is the relationship between responsibility (as a meta-virtue) and the particular virtues. I am not convinced that the case for responsibility to be considered as a virtue at all is well made. I wonder whether treating responsibility as a general quality of character or a moral disposition (and not a virtue) with various virtues (as discussed here) as components of responsibility would be more fitting. If the author insists on treating responsibility as a virtue, then the link between it and its sub-virtues should be properly substantiated.
There are a couple of very minor grammatical issues. For example, the sentence (lines 99-101) starting with 'As excellence...' needs revision.
Author Response
Thank you for your feedback!
I have addressed your comment regarding phronesis by adding a further explanatory sentence, as well as pointing out that I will return to the concept later on in the text. It now reads:
Practical wisdom is the virtuous person’s ability to judge the relative relevance of factors pertaining to a situation and weigh up these features in making moral decisions. It is the ability to translate the idea of moral good into concrete or practical situations. It can thus be understood as a purely good-willed cleverness, ruling how and when virtues are to be displayed [27]. I will further elaborate on practical wisdom in the following sections. (lines 160-163).
With regards to your more substantial comment, I have toned down that responsibility is to be understood as a virtue. I think it is valuable to point out that it can be understood as a virtue, but agree with you that it is not important for my argument that it is. It now reads:
Garrath Williams argues that responsibility should be understood as a good-making trait of character and defines it as “the readiness to respond to a plurality of normative demands” [9] (p. 469). Whether adhering to this understanding of responsibility as a virtue, or rather seeing responsibility as a quality of character or as a moral disposition, there is a value in trying to be more specific when discussing what it means to be responsible in a particular context; in this case subnational adaptation to climate change. (lines 59-64)
I have also changed the sentence where I called responsibility a meta-virtue. It now reads:
As mentioned, responsibility can be understood as a good-making characteristic or virtue. However, the purpose of this paper is to explore what kinds of virtues can be derived from an understanding of responsibility in the context of adaptation to sea level rise. Responsibility will therefore not be listed as a virtue in this paper. (lines 312-316)
In addition to this, I have also added a motivation of how planning can be understood as a profession and therefore a practice and made some minor language changes throughout the text. Hope that you find these changes satisfactory, and thank you again for your feedback!
Reviewer 2 Report
I enjoyed this article a lot. The application of virtue ethics to environmental issues has not been as prominent in environmental ethics as have deontological and utilitarian approaches. This article makes an important contribution to redressing that imbalance, and the specific application to the problem of adaptation to sea level rise provides relevant and practical suggestions for action.
Author Response
Thank you for your positive feedback! I have made a few changes in accordance with the other reviewer’s comments and also added a motivation of how planning can be understood as a profession.